# Kinesin-6 regulates cell-size-dependent spindle elongation velocity to keep mitosis duration constant in fission yeast

Lara Katharina Krüger[1]*, Jérémie-Luc Sanchez[1], Anne Paoletti[1], Phong Thanh Tran[1,2]*

[1]Institut Curie, PSL Research University, CNRS, UMR 144, Paris, France;
[2]Department of Cell and Developmental Biology, University of Pennsylvania, Philadelphia, United States

**Abstract** The length of the mitotic spindle scales with cell size in a wide range of organisms during embryonic development. Interestingly, in *C. elegans* embryos, this goes along with temporal regulation: larger cells speed up spindle assembly and elongation. We demonstrate that, similarly in fission yeast, spindle length and spindle dynamics adjust to cell size, which allows to keep mitosis duration constant. Since prolongation of mitosis was shown to affect cell viability, this may resemble a mechanism to regulate mitosis duration. We further reveal how the velocity of spindle elongation is regulated: coupled to cell size, the amount of kinesin-6 Klp9 molecules increases, resulting in an acceleration of spindle elongation in anaphase B. In addition, the number of Klp9 binding sites to microtubules increases overproportionally to Klp9 molecules, suggesting that molecular crowding inversely correlates to cell size and might have an impact on spindle elongation velocity control.

DOI: https://doi.org/10.7554/eLife.42182.001

**\*For correspondence:**
lara-katharina.kruger@curie.fr (LKK);
phong.tran@curie.fr (PTT)

**Competing interests:** The authors declare that no competing interests exist.

## Introduction

The size of the mitotic spindle robustly scales with cell size in various organisms. During early embryogenesis in *C. elegans*, *X. laevis* and various metazoans where cell size gradually decreases while the embryo undergoes successive rounds of cell division, spindle length can be reduced from 60 to a few micrometers (*Crowder et al., 2015*; *Hara and Kimura, 2009*; *Wühr et al., 2008*). Also apart from embryogenesis, spindle length has been shown to adjust to cell size in *S. cerevisiae* and human cells (*Rizk et al., 2014*; *Yang et al., 2016*). This relationship is regulated by the cytoplasmic volume through limiting cytoplasmic components, such as tubulin (*Good et al., 2013*; *Hazel et al., 2013*), as well as by molecules modulating microtubule dynamics (*Hara and Kimura, 2013*; *Lacroix et al., 2018*; *Reber and Goehring, 2015*; *Wilbur and Heald, 2013*). In general, the regulation of the size of subcellular structures is considered crucial for many cellular processes, and especially for mitosis. For instance, mitotic spindle length can ensure proper chromosome segregation. In *Drosophila* neuroblast mutant cells exhibiting abnormally long chromosome arms, cells elongate and form slightly longer spindles to exclude chromatid from the cleavage plane (*Kotadia et al., 2012*). Thus, in cells of different sizes the adjustment of spindle length might be critical to separate the two chromosome sets by an appropriate distance, avoiding that chromosomes intrude into the site of cell cleavage, which would result in chromosome cut (*Syrovatkina and Tran, 2015*).

Interestingly, evidence exists that such a scaling relationship is not restricted to size but also applies to the speed of mitotic processes. In *C. elegans* embryos, the velocity of spindle assembly in prophase and the velocity of spindle elongation in anaphase B adjust to cell size, such that longer spindles assemble and elongate with proportionally higher speeds (*Hara and Kimura, 2009*;

*Lacroix et al., 2018*). This may prevent extension of mitosis duration in larger cells. In fact, prolongation of mitosis has often been shown to result in cell death or arrest in subsequent cell cycle phases (*Araujo et al., 2016*; *Lanni and Jacks, 1998*; *Orth et al., 2012*; *Quignon et al., 2007*; *Rieder and Palazzo, 1992*; *Uetake and Sluder, 2010*). Thus, the time frame needed for chromosome segregation has to be regulated to ensure flawless cell division.

Still, it is not known how the scaling of spindle dynamics and cell size is established. Computer simulations suggest that the cell-size-dependent spindle elongation velocity in *C. elegans* embryos depends on the number of cortical force-generators pulling on spindle poles (*Hara and Kimura, 2009*). In contrast to this mechanism of anaphase B, many other organisms push spindle poles apart via microtubule sliding forces generated between antiparallel overlapping microtubules (MTs) at the spindle center (spindle midzone) (*Brust-Mascher et al., 2004*; *Brust-Mascher and Scholey, 2011*; *Hayashi et al., 2007*; *Khodjakov et al., 2004*; *Tolić-Nørrelykke et al., 2004*). In most organisms, these forces are generated by kinesin-5 (*Avunie-Masala et al., 2011*; *Brust-Mascher et al., 2009*; *Kapitein et al., 2008*; *Kapitein et al., 2005*; *Saunders et al., 1995*; *Sharp et al., 2000*) but also kinesin-6, as in fission yeast, can accomplish this task (*Fu et al., 2009*; *Rincon et al., 2017*). In vitro it has been shown that the forces generated by several kinesin-5 molecules, that crosslink and slide apart antiparallel overlapping microtubules add up (*Shimamoto et al., 2015*). However, regarding the complex structure of the mitotic spindle and the variety of forces potentially opposing outward sliding of microtubules, it is unclear if the production of a higher force by a bigger motor ensemble at the midzone would directly result in an increasing speed of spindle elongation.

By using the easily-genetically modifiable fission yeast *S. pombe*, we show that even though cells of various sizes adjust mitotic spindle length to cell size, the process of chromosome segregation is accomplished within a constant time frame. Independent of cell and spindle size, chromosomes are separated within approximately 30 min. To keep mitosis duration constant, larger cells speed up spindle elongation of their longer spindles. Focusing on anaphase B, we reveal that the kinesin-6 Klp9 (human CHO1/MKLP1) (*Nislow et al., 1992*; *Nislow et al., 1990*), which is a key regulator for anaphase B spindle elongation in fission yeast (*Fu et al., 2009*; *Rincon et al., 2017*), regulates spindle elongation velocity in a cell-size-dependent manner and consequently controls mitosis duration. We demonstrate that the velocity of spindle elongation is determined by the number of Klp9 molecules. With increasing cell size the motor amounts increase, leading to the recruitment of more motors to the spindle and an acceleration of spindle elongation. In addition, longer spindles form longer regions of antiparallel overlapping microtubules, within which more microtubules are assembled. Thereby, the number of Klp9 binding sites to microtubules increases to a greater extent than the number of Klp9 molecules on the spindle. The lower relative concentration of Klp9 motors at the midzone of longer spindles might additionally impact the regulation of spindle elongation velocity, based on the effects of molecular crowding. We propose a simple limited components model, where the amount of available Klp9 molecules and the amount of binding sites for the motor to the mitotic spindle regulates the velocity of spindle elongation and gives rise to a constant mitosis duration in cells of various sizes.

## Results

### Mitosis duration is constant in *S. pombe* cells, irrespective of spindle length or cell size

In the rod-shaped fission yeast, cell growth only occurs at cell tips and the diameter does not change (*Kelly and Nurse, 2011*). Final cell size can therefore be easily quantified by measuring cell length from tip-to-tip at mitosis, when cells stop growing. To investigate the impact of cell size on mitotic spindle length and mitosis duration, we used cell-cycle mutants, dividing at an abnormally short (*wee1-50*: 10.50 µm ± 1 µm) and abnormally long cell length (*cdc25-22*: 24.7 µm ± 2.6 µm) compared to wild-type cells (13.3 µm ± 0.7 µm) (*Figure 1A* and *Figure 1—source data 1*). Live cell imaging was performed in cells expressing α-tubulin (Atb2) linked to mCherry, to monitor the mitotic spindle, and the kinetochore component Mis12 tagged with GFP, to follow the position of chromosomes within the spindle (*Figure 1B*).

First, we determined if spindle length scales with cell size also in *S. pombe*. Maximum spindle length was measured just before spindle breakdown at the end of anaphase B by the length of the

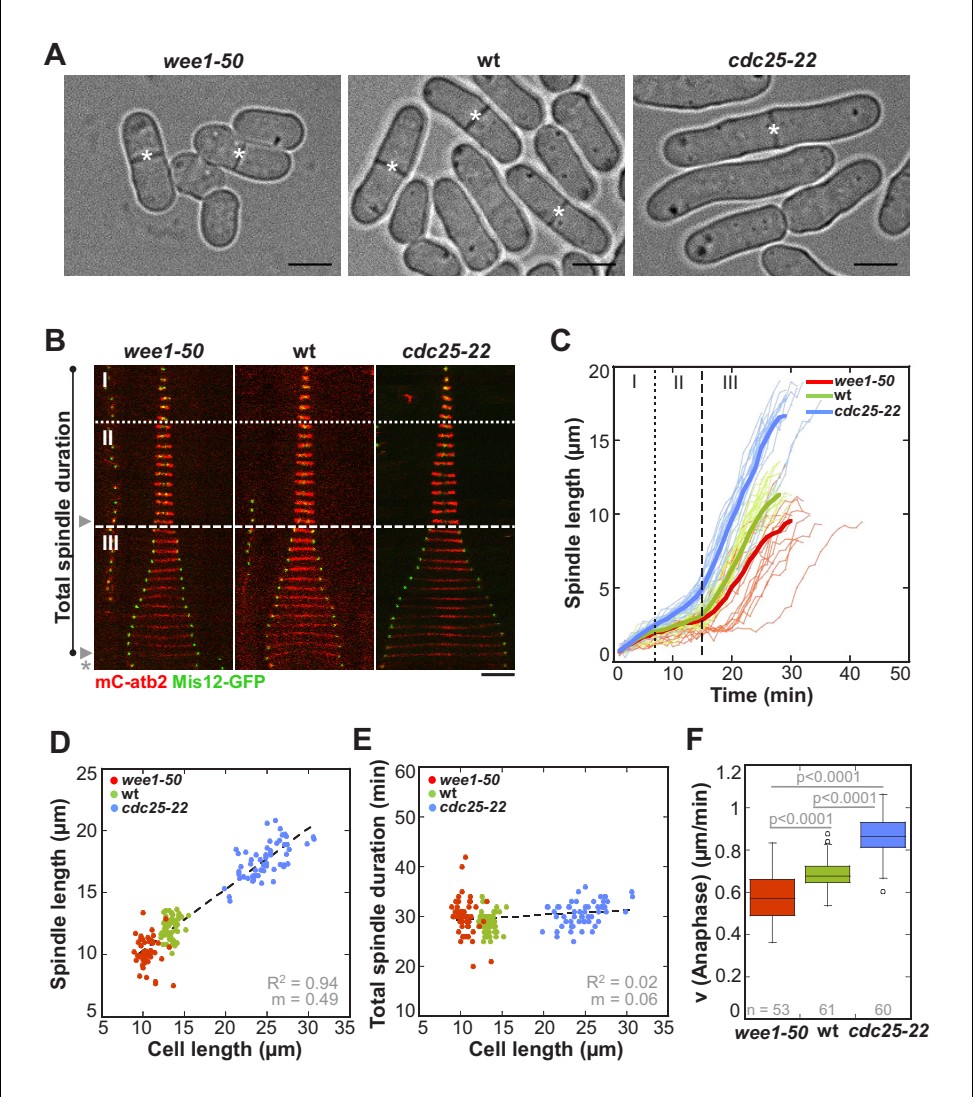

**Figure 1.** In *S. pombe* spindle length scales with cell size but total spindle duration is kept constant. (A) Brightfield images of *S. pombe* wild-type cells (central panel) and cell cycle mutants *wee1-50* (left panel) and *cdc25-22* (right panel). Cells reach their maximum cell length at mitosis/cell division, represented by the septum (asterisk). Scale bar, 5 μm. (B) Time-lapse images of *wee1-50*, wild-type (wt) and *cdc25-22* cells expressing mCherry-Atb2 (tubulin) and Mis12-GFP from spindle assembly to spindle breakdown (asterisk). Dotted line corresponds to transition from phase I to II (prophase to metaphase). Dashed line corresponds to transition from phase II to III (metaphase to anaphase). Arrowheads indicate the last time point of phase II or III, corresponding to spindle length plotted in (D) and *Figure 1—figure supplement 1*. Each frame corresponds to 1 min interval. Scale bar, 5 μm. (C) Comparative plot of spindle length dynamics of *wee1-50* (red curves, n = 20), wild-type (green curves, n = 20) and *cdc25-22* cells (blue curves, n = 20). Bold curves correspond to mean spindle dynamics. As in (B) dotted and dashed lines display phase transitions. (D) Final spindle length plotted against cell length (*wee1-50*: n = 53, wt: n = 61, *cdc25-22*: n = 60). (E) Total spindle duration plotted against cell length (*wee1-50*: n = 53, wt: n = 61, *cdc25-22*: n = 60). Data in (D–E) was fitted by linear regression (dashed lines), showing the regression coefficient $R^2$ and the slope m. (F) Boxplot comparison of anaphase B spindle elongation velocities (v) in *wee1-50*, wild-type and *cdc25-22* cells. Data from n cells was collected from three independent experiments. P values were calculated by Mann-Whitney U test.

DOI: https://doi.org/10.7554/eLife.42182.002

The following source data and figure supplements are available for figure 1:

**Source data 1.** Mean values of spindle length and dynamics.
DOI: https://doi.org/10.7554/eLife.42182.005

**Figure supplement 1.** Metaphase spindle length plotted against maximum cell length.
DOI: https://doi.org/10.7554/eLife.42182.003

*Figure 1 continued on next page*

*Figure 1 continued*

**Figure supplement 2.** Boxplot comparison of spindle elongation velocity in prophase (I), metaphase (II) and anaphase B (III).
DOI: https://doi.org/10.7554/eLife.42182.004

mCherry-Atb2 signal (*Figure 1B*: phase III-arrowhead). Short *wee1-50* cells assembled spindles with a maximum length of 10.14 ± 1.2 μm, wild-type cells of 11.93 ± 0.9 μm and long *cdc25-22* cells of 17.34 ± 1.4 μm on average (*Figure 1C and D*). This shows that maximum spindle length increased in proportion to cell size (*Figure 1D*: $R^2 = 0.94$, m = 0.49). Metaphase spindle length, measured at the end of metaphase (*Figure 1B*: phase II-arrowhead), also scaled with cell size, although to a slightly lower extent (*Figure 1—figure supplement 1*: $R^2 = 0.6$, m = 0.07). Hence, as observed in various other organisms, a strong correlation of spindle length and cell size exists in *S. pombe*.

We next analyzed spindle dynamics, that is the development of spindle length over time, in relation to cell size. Similar to wild-type, *wee1-50* and *cdc25-22* cells exhibited three easily distinguishable phases (*Figure 1B and C*): phase I with low spindle elongation velocity (prophase), phase II with no or little spindle elongation (metaphase and anaphase A) and phase III characterized by dramatic and comparatively fast spindle elongation (anaphase B). Strikingly, even though spindle length adjusted to cell length, duration of each phase did not correlate with cell size (*Figure 1B,C* and *Figure 1—source data 1*). Consequently, total spindle duration, that is the time from the appearance of the spindle until its breakdown (*Figure 1B*), was constant among cells of different sizes, taking place within approximately 30 min (*Figure 1E*). Total spindle duration, which we define as mitosis duration, was thus independent of cell size (*Figure 1E*: $R^2 = 0.02$, m = 0.06). This is achieved by an increase of the spindle elongation velocity with cell size. The scaling relationship of spindle dynamics and cell size was observed in all mitotic phases (*Figure 1—figure supplement 2*): the rate of spindle assembly in prophase, the velocity of very modest spindle elongation in metaphase and, most strikingly, the velocity of extensive spindle elongation in anaphase B increased with cell size (*Figure 1F*).

To test if these results indeed originate from changes in cell size and not from mutations of the cell cycle regulators wee1 and cdc25, we additionally used other methods to alter cell size without changing the basic levels of wee1 or cdc25 (*Figure 2*). Abnormally short cells were obtained by growing wild-type cells under conditions of starvation, which then divide at an average cell length of 10.91 ± 0.95 μm (*Figure 2A*: starved wt). Abnormally long cells were created by, first, treatment of wild-type cells with hydroxyurea (HU), which allows to arrest them in early S-Phase by disturbing DNA replication. After a few hours of incubation the drug can be washed out, allowing the cells to resume the cell cycle and enter mitosis at an average length of 15.73 ± 0.95 μm (*Figure 2A*: wt +HU). Second, we used an analogue-sensitive (as) mutant of cdc2, homolog of the cyclin-dependent kinase Cdk1 (*Figure 2A*: *cdc2-asM17*). Treatment of this mutant with ATP-analogue inhibitors blocks cells at the G1/S and G2/M transition due to inactivation of cdc2. After washout of the inhibitor, full function of cdc2 can be restored and cells enter mitosis (*Aoi et al., 2014*). Depending on the duration of incubation with the ATP-analogue inhibitor NM-PP1, *cdc2-asM17* cells divided at cell lengths ranging from 12.24 to 27.09 μm (*Figure 2A*: *cdc2-asM17* + NMPP1). Using these methods, we could confirm the previously observed scaling relationship of spindle length and cell length (*Figure 2B*), the constant mitosis duration in cells of varying sizes (*Figure 2C*), as well as the increase of spindle elongation velocity with increasing cell size (*Figure 2D*). Thus, these cells behave similarly compared to the *wee1-50* and *cdc25-22* mutants, indicating that the observed changes of spindle dynamics stem from differences in cell size, not different levels of cell cycle regulators.

Taken together, in fission yeast spindle length scales with cell size, however mitosis can still be realized within a constant time frame, through a cell-size-dependent spindle elongation velocity.

## Kinesin-6 Klp9 controls spindle elongation velocity and mitosis duration

We set out to understand how spindle dynamics adjust to cell size. The most significant cell-size-dependent acceleration of spindle elongation occurred in anaphase B (*Figure 1—figure supplement 2*), which takes approximately half of the total mitosis time in fission yeast (*Figure 1B and C*). Therefore, we focused on this mitotic phase to examine how mitosis duration is maintained constant, irrespective of cell size.

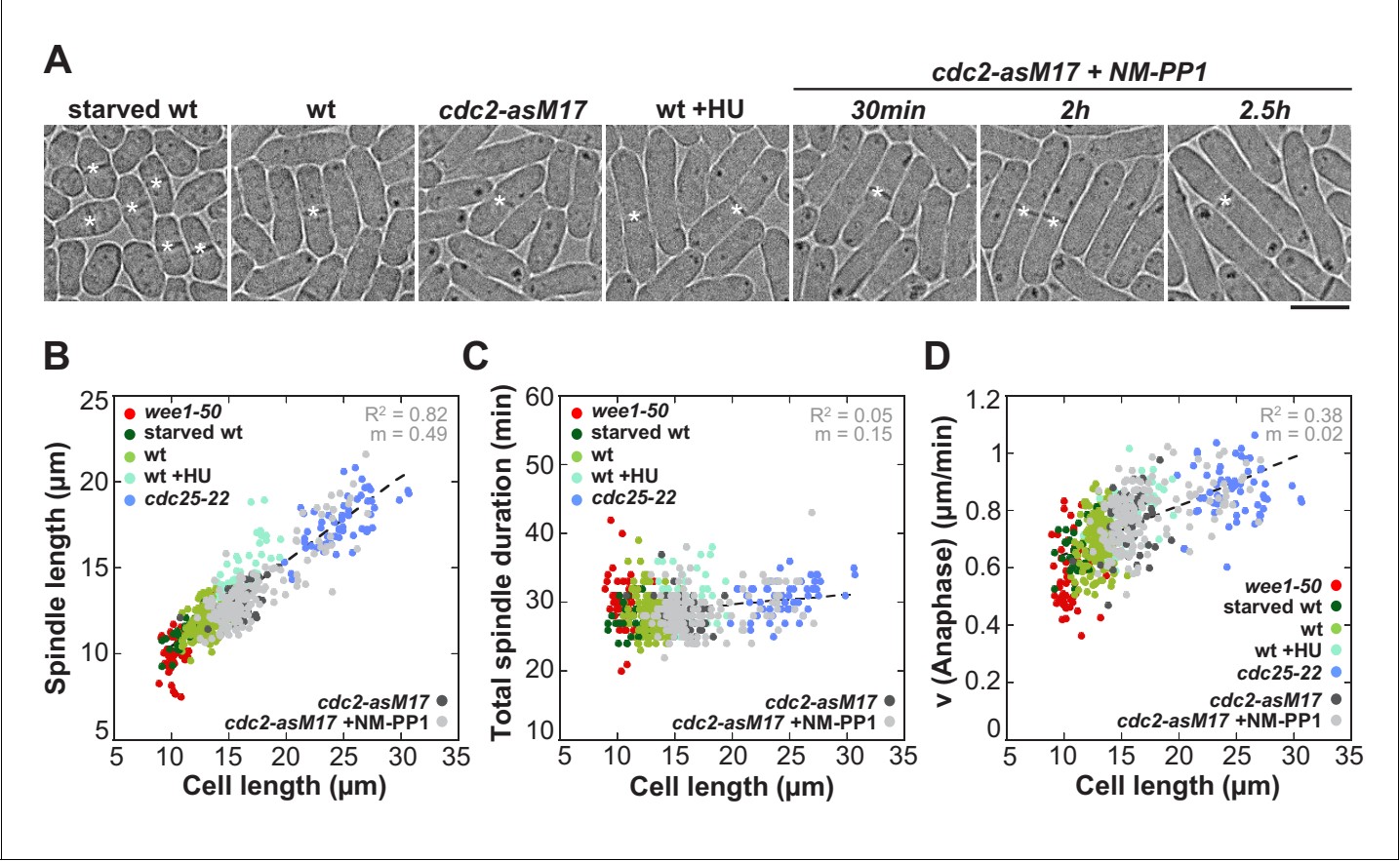

**Figure 2.** Spindle length and spindle elongation velocity scales with cell length in multiple conditions. (**A**) Brightfield images of *S. pombe* wild-type cells in starvation (starved wt), wild-type cells in exponential growth phase (wt), *cdc2-asM17* cells, wild-type cells treated with hydroxyurea (wt+HU), *cdc2-asM17* cells treated with NM-PP1 for 30 min, 2 hr or 2.5 hr. Cells reach their maximum cell length at mitosis/cell division, represented by the septum (asterisk). Scale bar, 5 µm. (**B**) Final spindle length plotted against cell length. (**C**) Total spindle duration plotted against cell length. (**D**) Anaphase spindle elongation velocity plotted against cell length. Data in (**B–D**) was fitted by linear regression (dashed lines), showing the regression coefficient $R^2$ and the slope m. Data was obtained from n cells (starved wt: n = 27, wt: n = 133, wt +HU: n = 52, *cdc2-asM17*: n = 49, *cdc2-asM17* + NM-PP1: n = 163) was collected from three independent experiments.

DOI: https://doi.org/10.7554/eLife.42182.006

In *S. pombe*, outward sliding of antiparallel microtubules is predominantly executed by the kinesin-6 Klp9 (*Figure 3C*) (*Fu et al., 2009*). At anaphase onset the highly conserved MT-crosslinker Ase1 (PRC1/MAP65) bundles antiparallel microtubules, organizing a structure of overlapping microtubules that is called spindle midzone (*Bieling et al., 2010*; *Gaillard et al., 2008*; *Glotzer, 2009*; *Janson et al., 2007*; *Loïodice et al., 2005*; *Mollinari et al., 2002*; *Pellman et al., 1995*; *Schuyler et al., 2003*; *Yamashita et al., 2005*). In this position, Ase1 is thought to recruit Klp9, which is localized to the nucleus, in a dephosphorylation-dependent manner to the midzone (*Figure 3B and C*) (*Fu et al., 2009*). As a plus-end directed homotetrameric motor, Klp9 can slide apart the antiparallel microtubules and elongate the mitotic spindle (*Fu et al., 2009*; *Rincon et al., 2017*).

Deletion of Klp9 decreased the speed of anaphase B spindle elongation velocity in *wee1-50*, wild-type and *cdc25-22* cells (*Figure 3A*). Strikingly, anaphase spindle elongation velocities decreased to very similar values in all three cell types (*Figure 3D and E*), as well as in the *cdc2-asM17* mutant (*Figure 3—figure supplement 1*), ranging between 0.3–0.5 µm/min. Thus, the correlation between spindle elongation velocity and cell size is abolished in absence of Klp9. Despite this, the scaling relationship of spindle length and cell size was still effective (*Figure 3F* and *Figure 3—figure supplement 2*). As a result of similar spindle elongation velocities in the absence of Klp9, longer cells needed proportionally more time to complete chromosome segregation compared to

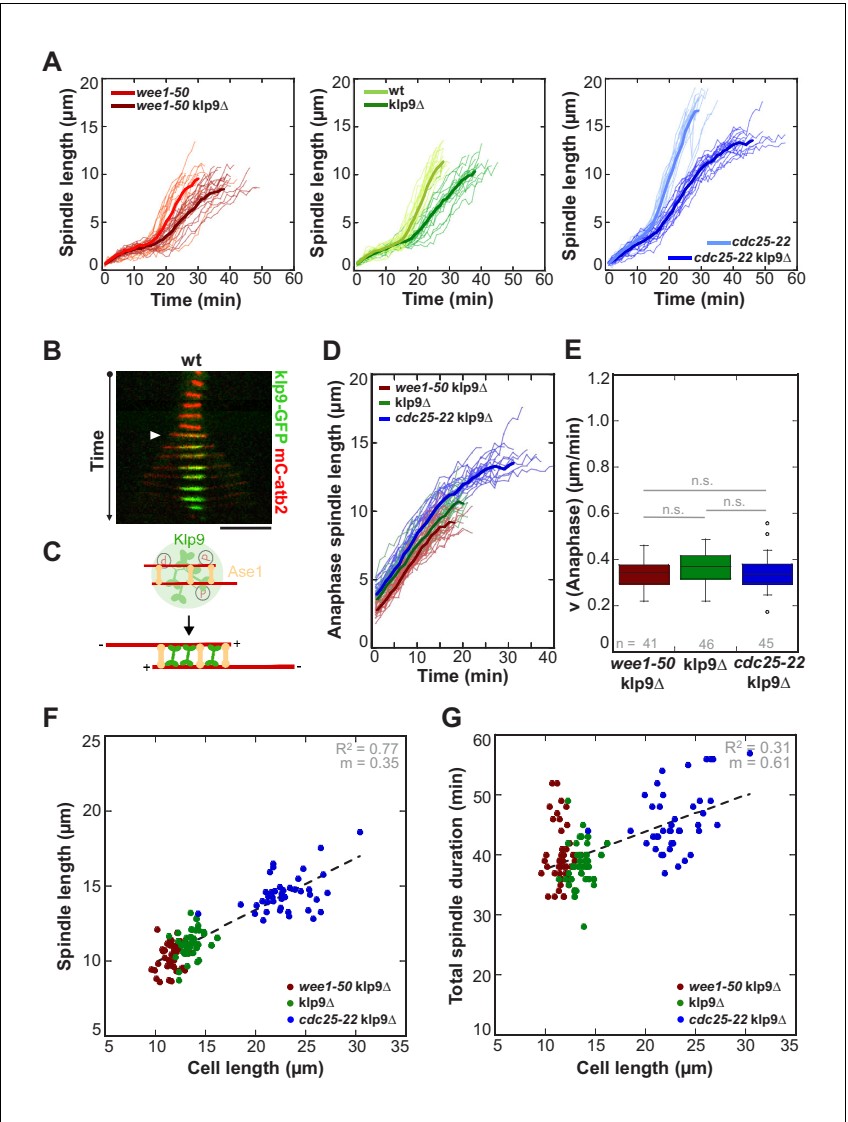

**Figure 3.** Kinesin-6 Klp9 deletion abolishes cell-size-dependent spindle elongation velocity and constant mitosis duration. (**A**) Comparative plots of spindle length dynamics of *wee1-50* (light-red, n = 20) and *wee1-50*klp9Δ (dark-red, n = 20) [left panel], wild-type (light-green, n = 20) and klp9Δ (dark-green, n = 20) [central panel], *cdc25-22* (light-blue n = 20) and *cdc25-22*klp9Δ (dark-blue, n = 20) [right panel]. Bold curves correspond to mean spindle dynamics of each cell type. (**B**) Time-lapse image from mitosis onset until spindle breakdown of a wild-type (wt) cell expressing mCherry-Atb2 (tubulin) and Klp9-GFP. Each frame corresponds to 2 min interval. Arrowhead marks the start of anaphase B. Scale bar, 5 μm. (**C**) Model for anaphase spindle elongation by Klp9, based on *Fu et al., 2009*. Microtubules are shown in red, Klp9 in green and Ase1 in orange. (**D**) Comparative plot of anaphase spindle dynamics of *wee1-50*klp9Δ (dark-red, n = 20), klp9Δ (dark-green, n = 20) and *cdc25-22*klp9Δ (dark-blue, n = 20). Bold curves correspond to mean spindle dynamics. (**E**) Box plot comparison of anaphase B spindle elongation velocities (v) in *wee1-50*klp9Δ, klp9Δ and *cdc25-22*klp9Δ. P values were calculated by Mann-Whitney U test; data sets are defined as not significantly different (n.s.) if p>0.05. (**F**) Final spindle length in the klp9Δ background plotted against maximum cell length (*wee1-50*klp9Δ: n = 41, klp9Δ: n = 46, *cdc25-22*klp9Δ: n = 45). (**G**) Total spindle duration in the klp9Δ background plotted against cell length (*wee1-50*klp9Δ: n = 41, klp9Δ: n = 46, *cdc25-22*klp9Δ: n = 45). Data in (**F–G**) was fitted by linear regression (dashed lines), showing the regression coefficient $R^2$ and the slope m. Data obtained from n cells was collected from three independent experiments.

DOI: https://doi.org/10.7554/eLife.42182.007

The following figure supplements are available for figure 3:

**Figure supplement 1.** Anaphase spindle elongation velocity plotted against cell length of *cdc2-asM17*klp9Δ and *cdc2-asM17*klp9Δ + NM-PP1.

*Figure 3 continued on next page*

*Figure 3 continued*

DOI: https://doi.org/10.7554/eLife.42182.008

**Figure supplement 2.** Final spindle length plotted against cell length of *cdc2-asM17*klp9Δ and *cdc2-asM17*klp9Δ + NM-PP1.

DOI: https://doi.org/10.7554/eLife.42182.009

**Figure supplement 3.** Total spindle duration plotted against cell length of *cdc2-asM17*klp9Δ and *cdc2-asM17*klp9Δ + NM-PP1.

DOI: https://doi.org/10.7554/eLife.42182.010

shorter cells (*Figure 3G* and *Figure 3—figure supplement 3*). Thus, the phenomenon of constant mitosis duration irrespective of cell size is abolished in absence of Klp9. Together, these results suggest that the kinesin-6 Klp9 is a key component of the mechanism that adjusts spindle elongation velocity to cell size.

## The number of Klp9 molecules determines spindle elongation velocity

It was previously suggested by computer simulations that the number of force generators acting on the spindle could regulate the speed of spindle elongation (*Hara and Kimura, 2009*). To test this hypothesis, we performed intensity measurements of Klp9-GFP expressed in *wee1-50*, wild-type and *cdc25-22* cells. Following the Klp9-GFP signal intensity throughout anaphase B (*Figure 4A and B*), revealed that, despite fluctuations, the intensity was largely unaffected from shortly after anaphase onset (*Figure 4A and B*: arrowhead) to the end of anaphase B spindle elongation, before the signal intensity dropped down when the spindle disassembled (*Figure 4A and B*: asterisk). In general, the Klp9-GFP intensity increased from *wee1-50* to wild-type to *cdc25-22* cells at every given time point (*Figure 4B*). Moreover, in addition to the Klp9-GFP intensity at the midzone of anaphase spindles (*Figure 4C*: unfilled dots) also the total Klp9-GFP intensity in cells (*Figure 4C*: filled dots), measured shortly before mitosis onset within the nucleus, increased with cell size (*Figure 4C* and *Figure 4—source data 1*). This was similarly observed in the *cdc2as-M17* mutant (*Figure 4—figure supplement 1*), showing that the increased recruitment of Klp9 to the spindle occurs due to an increase in cell size, and is not a result of mutations of *wee1* and *cdc25*.

It is tempting to think that total numbers of Klp9 molecules could directly determine the extent of Klp9 recruitment to the spindle. Klp9 could thus represent a limited component within the regulation of spindle elongation velocity, such as tubulin does within the regulation of spindle length (*Good et al., 2013*; *Hazel et al., 2013*). In this classical model for scaling relationships, the concentration of a limited component, is constant and thus total amounts increase proportional to cell size. Consequently, in larger cells more of this limited component is available, for example leading to the assembly of longer mitotic spindles due to the presence of more tubulin (*Reber and Goehring, 2015*) or potentially an acceleration of spindle elongation due to the presence of more Klp9 motors, allowing the recruitment of more motors to the spindle.

The latter is supported by the fact that the ratio of Klp9-GFP intensity at anaphase spindles over its total intensity in the cell, referred to as Klp9 spindle fraction, did not correlate with cell size (*Figure 4D*: $R^2$ = 0.08, m = 0.0015; and *Figure 4—figure supplement 2*). More precisely, approximately 40% of the total cellular Klp9 pool was recruited to anaphase spindles in all cell types (*Figure 4D* and *Figure 4—figure supplement 2*). This indicates that the number of motors recruited to the spindle directly correlates with the total number of available motors. Moreover, if the extent of Klp9 recruitment is similar in cells of different sizes, we expect the Klp9 concentration in the cell to be constant as well. The concentration of Klp9 was estimated by the ratio of total intensity over cell volume, with the cell volume being calculated by treating the rod-shaped cell as a cylinder (*Figure 4—figure supplement 3*). Indeed, we found the Klp9 concentration in the cell to be largely independent of cell size (*Figure 4E*: $R^2$ = 0.09, m = −0.02 and *Figure 4—source data 1*).

Similarly to the quantities of Klp9, we observed, that the amount of tubulin, either the total amount of tubulin within the cell (*Figure 4—figure supplement 4*: filled dots) or the amount of tubulin incorporated within the late anaphase spindle (*Figure 4—figure supplement 4*: unfilled dots) increased proportional to cell size. Furthermore the concentration of tubulin in the cell, estimated by the ratio of total mCherry-Atb2 intensity and cell volume, remained constant in cells of different sizes (*Figure 4—figure supplement 5*).

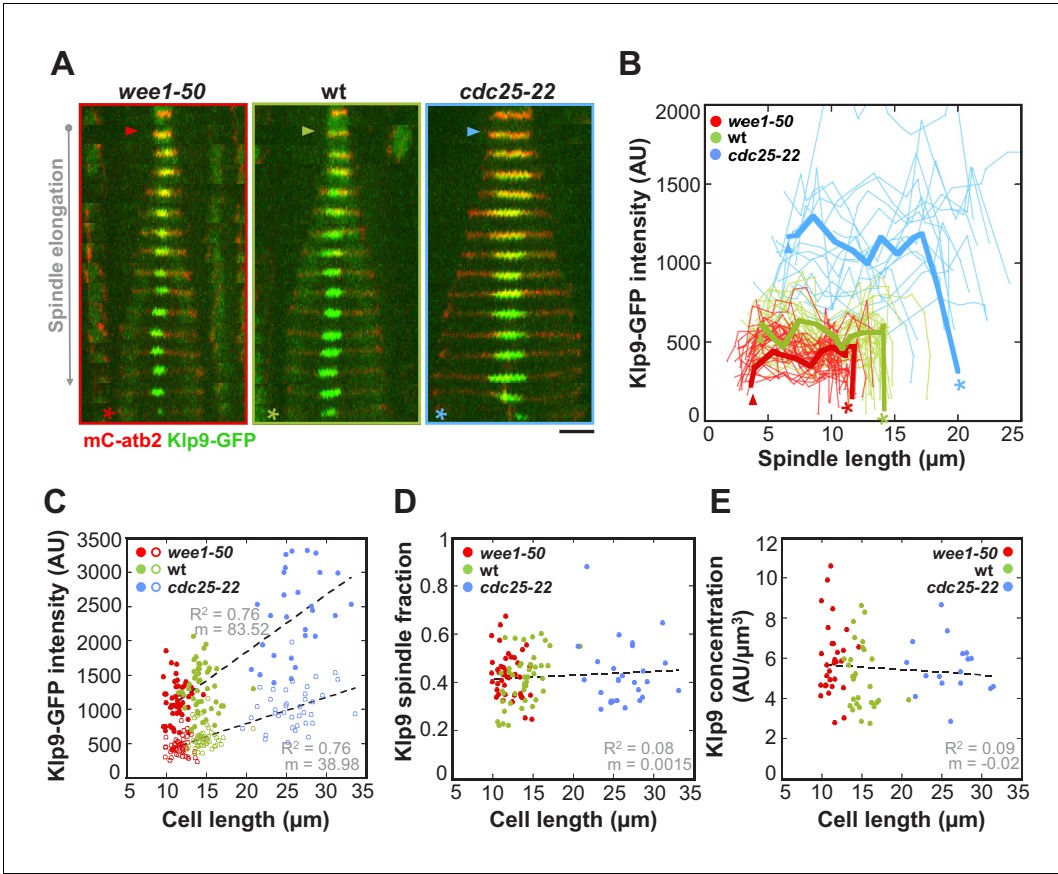

**Figure 4.** The total number of Klp9 molecules and the number of motors on anaphase spindles increase with cell size. (**A**) Time-lapse images of *wee1-50*, wild-type (wt) and *cdc25-22* cells expressing mCherry-Atb2 (tubulin) and Klp9-GFP from anaphase onset (arrowhead) to spindle breakdown (asterisk). Each frame corresponds to 1 min interval. Scale bar, 5 µm. (**B**) Comparative plot of Klp9-GFP intensity over spindle length. Each line corresponds to the Klp9-GFP intensity values of one cell throughout progressing spindle elongation in anaphase. Bold curves correspond to a representative cell of each cell type. (**C**) Total Klp9-GFP intensity (•, filled dots) and Klp9-GFP intensity at anaphase spindles (○, unfilled dots) plotted against cell length (*wee1-50*: n = 48, wt: n = 46, *cdc25-22*: n = 30). Shown values of Klp9-GFP intensity of anaphase spindles correspond to the mean intensity values of late anaphase. (**D**) Ratio between Klp9-GFP intensity at the anaphase spindle and the total Klp9-GFP intensity, referred to as Klp9 spindle fraction, plotted against cell length (*wee1-50*: n = 48, wt: n = 46, *cdc25-22*: n = 30). (**E**) Ratio of total Klp9-GFP intensity and cell volume plotted against cell length (*wee1-50*: n = 30, wt: n = 29, *cdc25-22*: n = 17). Data in (**C–E**) was fitted by linear regression (dashed lines), showing the regression coefficient $R^2$ and the slope m. Data obtained from n cells was collected from three independent experiments.

DOI: https://doi.org/10.7554/eLife.42182.011

The following source data and figure supplements are available for figure 4:

**Source data 1.** Mean values of Klp9-GFP intensity and concentration.
DOI: https://doi.org/10.7554/eLife.42182.017

**Figure supplement 1.** Total Klp9-GFP intensity and Klp9-GFP intensity at anaphase spindles plotted against cell length of *cdc2-asM17* and *cdc2-asM17* + NM-PP1.
DOI: https://doi.org/10.7554/eLife.42182.012

**Figure supplement 2.** Ratio between Klp9-GFP intensity at anaphase spindles and in the nucleus plotted against cell length of *cdc2-asM17* and *cdc2-asM17* + NM-PP1.
DOI: https://doi.org/10.7554/eLife.42182.013

**Figure supplement 3.** Estimation of cell volume of *S. pombe* cells.
DOI: https://doi.org/10.7554/eLife.42182.014

**Figure supplement 4.** Total mCherry-Atb2 intensity and mCherry-Atb2 intensity of anaphase spindles in *wee1-50*, wild-type and *cdc25-22* cells.
DOI: https://doi.org/10.7554/eLife.42182.015

*Figure 4 continued on next page*

*Figure 4 continued*

**Figure supplement 5.** Ratio of total mCherry-Atb2 intensity and cell volume plotted against cell length.
DOI: https://doi.org/10.7554/eLife.42182.016

To further probe the impact of the amount of Klp9 molecules on spindle elongation velocity, we modified expression levels of *klp9* by inserting the thiamine repressible *nmt* promoters *pnmt1*, *pnmt41 or pnmt81* upstream of the endogenous *klp9* open reading frame. Expression of *klp9* under the control of *pnmt81* led to low, *pnmt41* to mild and *pnmt1* to very strong overexpression (*Figure 5A*). Spindle dynamics were measured in cells overexpressing untagged *klp9* and intensity measurements were performed in cells expressing *GFP-klp9*. Anaphase B spindle elongation velocity was altered in a dose-dependent manner relative to the expression level of *klp9* (*Figure 5B*). Generally, the stronger *klp9* was overexpressed, the faster anaphase spindles elongated (*Figure 5C*). We further tested if the acceleration of spindle elongation was due to an increase in Klp9 recruitment to the spindle. Intensity measurements in cells expressing *GFP-klp9* under the control of the different *nmt* promoters revealed an increase of GFP-klp9 intensities, measured in anaphase spindles, in proportion to expression levels (*Figure 5E* and *Figure 5—source data 1*). An increase of GFP-klp9 signal on the spindle correlates with an increase of spindle elongation velocity until a certain upper-limit (*Figure 5D*). At a velocity of approximately 1.1–1.2 µm/min no strong acceleration occurred even though GFP-klp9 intensities increased further (*Figure 5D*). This upper-limit could stem from either the maximum velocity a Klp9 motor can walk, the saturation of binding sites for Klp9 or the velocity of microtubule growth.

However, we observed a difference in the slope of the velocity versus intensity correlation between the control condition (*wee1-50*, wt and *cdc25-22*) and cells in which *klp9* was overexpressed. Linear regression of mean values of control cells gives a greater slope ($m = 5.2 \times 10^{-3}$) than cells in which *klp9* was overexpressed ($m = 0.3 \times 10^{-3}$; values used for GFP-klp9 <500 AU) (*Figure 5—figure supplement 1*). This indicates that other factors might contribute to ensure such an efficient adjustment of spindle elongation to cell size, as accomplished in the control condition. Nevertheless, modifying the number of Klp9 molecules is sufficient to regulate spindle elongation velocity.

## Ase1, Cut7 and Klp2 do not regulate cell-size-dependent spindle elongation velocity

How is the cell-size-dependent spindle elongation velocity regulated besides the available number of kinesin-6 molecules? Many other midzone components assist or oppose microtubule sliding (*Scholey et al., 2016*) and could thus regulate spindle elongation velocity as well. However, since the scaling relationship with cell size was completely abolished in absence of Klp9, we argue that, this additional level of regulation has to occur through modifying the action of Klp9.

Interaction of Klp9 with the MT crosslinker Ase1 (*Figure 6G*) has been proposed to bring Klp9 into a tetrameric conformation at the spindle midzone, in which it preferentially binds antiparallel microtubules, thus enhancing the efficiency of spindle elongation (*Fu et al., 2009*). Therefore, we decided to examine the role of Ase1 within the underlying mechanism. Ase1 localizes to the midzone shortly before anaphase onset (*Figure 6A*: arrowhead) and stays at the midzone as well as at spindle poles throughout anaphase B, as previously reported (*Loïodice et al., 2005*; *Yamashita et al., 2005*). Intensity measurements in wild-type, *wee1-50* and *cdc25-22* cells expressing Ase1-GFP revealed an increase at the spindle midzone proportional to cell size (*Figure 6B,C* and *Figure 6—source data 1*). To probe the role of the MT bundler, we overexpressed *ase1* by using *nmt* promoters. Expression of *ase1* under the control of *pnmt81* and *pnmt41* decreased anaphase B spindle elongation velocity in a dose-dependent manner (*Figure 6D and E*). Cells, which expressed *ase1* under the control of the strongest *nmt* promoter (*pnmt1*) appeared very sick and spindles did not seem to elongate at all after metaphase (data not shown). Thus, if present in excess, Ase1 resists spindle elongation. This is consistent with previous in vitro studies where Ase1 has been shown to act as a brake, opposing the action of kinesins, which slide apart antiparallel microtubules (*Braun et al., 2011*; *Lansky et al., 2015*). We thus wondered if Ase1 instead of promoting spindle elongation, opposes the action of Klp9. To test this, we reduced the amount of Ase1 by using an Ase1 shut-off strain (*ase1^Off*). In contrast to the full deletion of *ase1*, the presence of a reduced

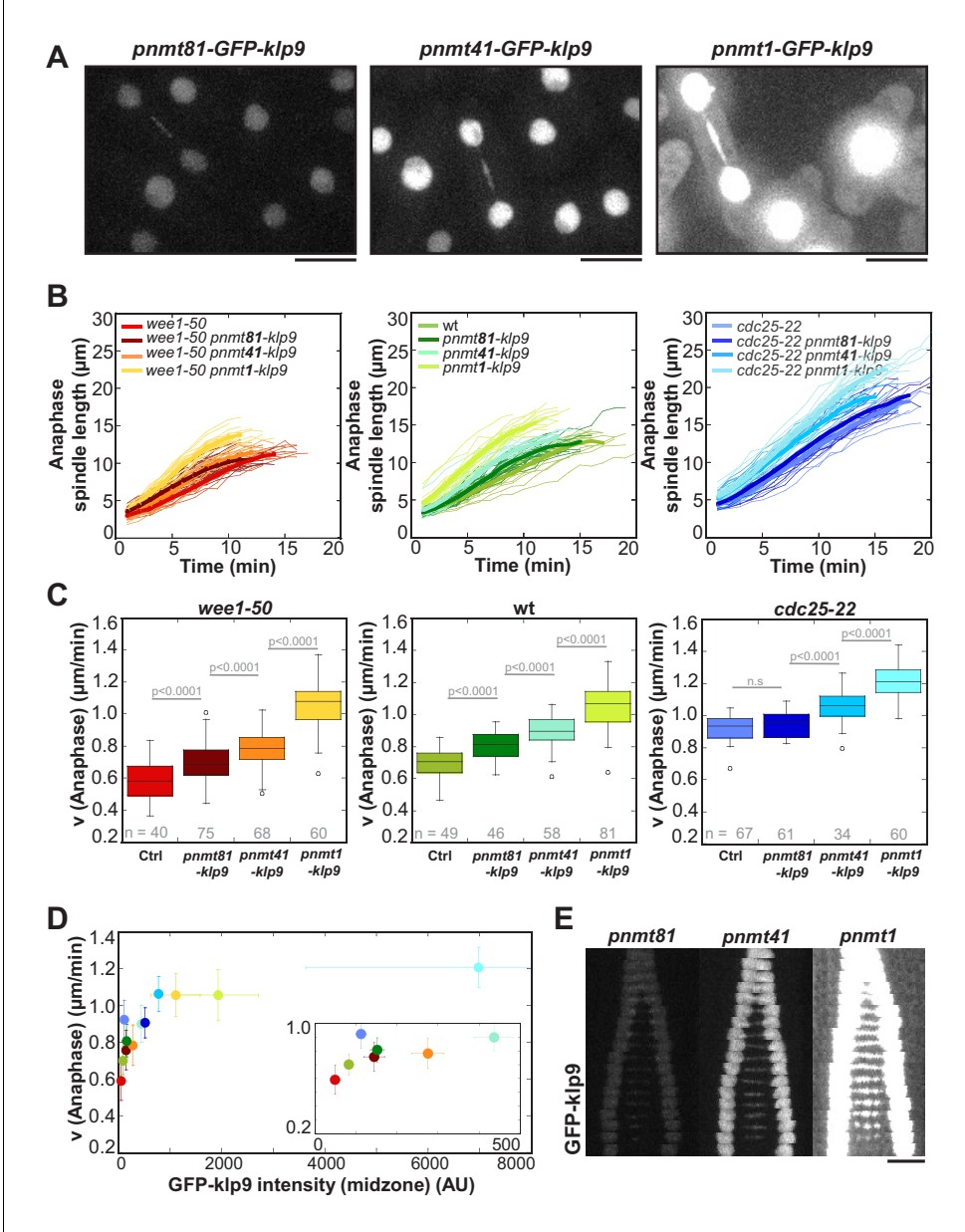

**Figure 5.** Overexpression of Klp9 accelerates anaphase B spindle elongation in a dose-dependent manner. (**A**) Images of cells expressing *GFP-klp9* under the control of thiamine repressible *nmt* promoters. Scale bar, 5 μm. (**B**) Comparative plot of spindle length dynamics of *wee1-50* (left panel), wild-type (middle panel) and *cdc25-22* cells (right panel) with endogenous *klp9* expression, *pnmt81-klp9*, *pnmt41-klp9* and *pnmt1-klp9* (each strain: n = 20). (**C**) Box plot comparison of anaphase B spindle elongation velocities (v) in cells with endogenous *klp9* expression (Ctrl), *pnmt81-klp9*, *pnmt41-klp9* and *pnmt1-klp9*. P values were calculated by Mann-Whitney U test. (**D**) Mean values and corresponding standard deviations of anaphase B velocity plotted against mean values of GFP-klp9 intensity at anaphase spindles. The inset shows the same data on a 0 to 500 AU x-axis and a 0.2 to 1.0 μm/min y-axis. The color code is equal to the one used in (**B**) and (**C**). (**E**) Time-lapse images from metaphase to spindle breakdown in cells expressing GFP-klp9 under the control of *nmt* promoters. Each frame corresponds to 1 min interval. Scale bar, 5 μm. Data obtained from n cells was collected from three independent experiments.

DOI: https://doi.org/10.7554/eLife.42182.018

The following source data and figure supplement are available for figure 5:

**Source data 1.** Mean values of spindle elongation velocity and the Klp9-GFP intensity at anaphase spindles in pnmt-klp9 strains.

DOI: https://doi.org/10.7554/eLife.42182.020

*Figure 5 continued on next page*

*Figure 5 continued*

**Figure supplement 1.** Linear regression of the function of spindle elongation velocity and Klp9-GFP intensity.
DOI: https://doi.org/10.7554/eLife.42182.019

amount of the MT-bundler still allows the formation of relatively stable spindles, since they do not break early in anaphase B (*Figure 6—figure supplement 1*). We reasoned that if Ase1 acts as a brake for MT sliding, this strong reduction of the Ase1 pool would result in an acceleration of spindle elongation. However, we could not observe a significant difference in anaphase B spindle elongation between wild-type and *ase1^off* cells (*Figure 6F and H*), even though Ase1-GFP intensities at the midzone were strongly diminished (*Figure 6—figure supplement 2*). Accordingly, analysis of Klp9-GFP signals in both conditions did not reveal significant differences in intensity of Klp9-GFP or signal length at the midzone (*Figure 6—figure supplements 3* and *4*). In some cells the distribution of Klp9 over the spindle was expanded, as previously observed upon *ase1* deletion (*Fu et al., 2009*). Nevertheless, even though only a few Ase1 molecules organize the midzone, Klp9 can be normally recruited and slide apart antiparallel spindle MTs at wild-type speed. Together, with regards to velocity, Ase1 does not promote or oppose spindle elongation in anaphase when expressed at endogenous levels, even though it has been shown to be essential for spindle stability (*Janson et al., 2007*; *Loïodice et al., 2005*; *Yamashita et al., 2005*). The deceleration of spindle elongation upon Ase1 overexpression might only occur when Ase1 is strongly accumulated but not under physiological conditions. Accordingly, in vitro microtubule sliding is stalled only upon a Ase1: Ncd (*Drosophila* kinesin-14) ratio of 4:1 (*Braun et al., 2011*), or a ratio of 25:1 using PRC1 and Eg5 (*Subramanian et al., 2010*).

Besides Ase1, Kinesin-5 (*Figure 6G*) could oppose the action of Klp9 and thus modify spindle elongation velocity. For instance, in absence of the *C. elegans* kinesin-5 BMK-1 spindle elongation during anaphase is abnormally fast (*Saunders et al., 2007*). In fission yeast, as in most other organisms, deletion of the kinesin-5 Cut7 is lethal, due to the formation of monopolar spindles. However, upon deletion of the kinesin-14 Pkl1, Cut7 deletion is viable and cells are able to form bipolar spindles (*Olmsted et al., 2014*; *Rincon et al., 2017*; *Syrovatkina and Tran, 2015*; *Yukawa et al., 2018*). We therefore examined spindle dynamics in a double pkl1Δcut7Δ deletion strain. Comparing anaphase velocity with wild-type cells showed a slight but significant decrease (*Figure 6H*). Since the absence of Pkl1 alone does not affect elongation velocity (*Figure 6H*), this decrease may result from the absence of Cut7. Therefore, Cut7 does not act as a brake in fission yeast and might in the opposite be required to promote spindle elongation in Anaphase B, to a small extent, as previously suggested (*Rincon et al., 2017*). However, even with its role in promoting spindle elongation, Cut7 is not sufficient to regulate a cell-size-dependent spindle elongation velocity, since the velocity of spindle elongation is similar in cells of different size in absence of Klp9 but presence of Cut7 (klp9Δ strains, *Figure 3*).

Another candidate was the kinesin-14 Klp2, a minus-end directed motor present at the midzone (*Figure 6G*) which was recently shown to generate inward forces against Cut7 in fission yeast (*Yukawa et al., 2018*). However, its deletion did not significantly alter anaphase B elongation velocity either (*Figure 6H*).

Together, even though Ase1 interacts with Klp9 (*Fu et al., 2009*), Cut7 contributes to microtubule sliding in anaphase in absence of Klp9 (*Rincon et al., 2017*), and Klp2 opposes outward sliding of microtubules by Cut7 (*Yukawa et al., 2018*), these spindle components do not seem to have a substantial impact on the regulation of the cell-size-dependent spindle elongation velocity.

## Microtubule density and overlap length increase with spindle length

The spindle midzone is a key structure for anaphase B spindle elongation (*Khodjakov et al., 2004*; *Scholey et al., 2016*; *Tolić-Nørrelykke et al., 2004*) and the site of action for Klp9 (*Fu et al., 2009*). We hypothesized that differences in its structure could therefore affect spindle elongation velocity in addition to the number of Klp9 molecules.

Line scan-analysis of Klp9-GFP signals at anaphase spindles in *wee1-50*, wild-type and *cdc25-22* cells revealed an increase in signal length in *cdc25-22* cells compared to wild-type and *wee1-50* cells (*Figure 7A*). This indicates, that the length of microtubule overlap increases in longer spindles.

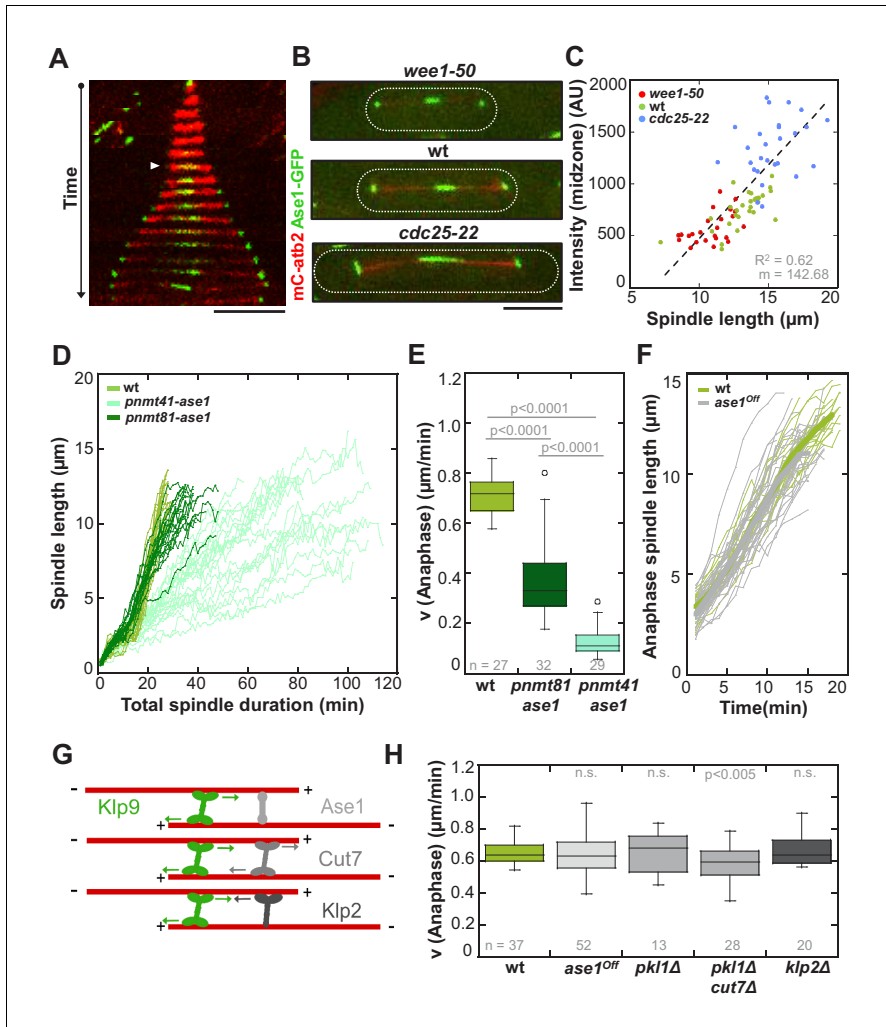

**Figure 6.** Ase1, Cut7 and Klp2 do not contribute to the regulation of cell-size-dependent spindle elongation velocity. (**A**) Time-lapse image from mitosis onset until spindle breakdown of a wild-type cell expressing mCherry-Atb2 (tubulin) and Ase1-GFP. Arrowhead corresponds to anaphase B onset. Each frame corresponds to 2 min interval. (**B**) Images of *wee1-50*, wild-type (wt) and *cdc25-22* cells expressing mCherry-Atb2 (tubulin) and Ase1-GFP in late anaphase. Scale bar, 5 μm. (**C**) Ase1-GFP intensity at anaphase spindles plotted against final spindle length (*wee1-50*: n = 24, wt: n = 28, *cdc25-22*: n = 30). Data was fitted by linear regression (dashed lines), showing the regression coefficient R² and the slope m. (**D**) Comparative plot of spindle length dynamics of wild-type (n = 27) cells, *pnmt81-ase1* (n = 32) and *pnmt41-ase1* (n = 29). (**E**) Box plot comparison of anaphase B spindle elongation velocities (v) in wild-type, *pnmt81-ase1* and *pnmt41-ase1* cells. (**F**) Comparative plot of anaphase spindle dynamics of wild-type (n = 28) and *ase1^Off* cells (n = 52). Bold curves correspond to mean spindle dynamics. (**G**) Model of the role of tested midzone components. (**H**) Box plot comparison of anaphase B spindle elongation velocities (v) in wild-type cells, ase1-Shut-Off (*ase1^Off*), *pkl1Δ*, the double-deletion *pkl1Δcut7Δ* and *klp2Δ* cells. P values were calculated by Mann-Whitney U test; data sets are defined as not significantly different (n.s.) if p>0.05. Data obtained from n cells was collected from three independent experiments.

DOI: https://doi.org/10.7554/eLife.42182.021

The following source data and figure supplements are available for figure 6:

**Source data 1.** Mean values of Ase1-GFP intensity and signal length.
DOI: https://doi.org/10.7554/eLife.42182.026
**Figure supplement 1.** Upon reduction of the Ase1 level spindles remain stable until late anaphase.
DOI: https://doi.org/10.7554/eLife.42182.022
**Figure supplement 2.** GFP-Ase1 intensity at anapahse spindles in *ase1^Off* and wild-type cells.
DOI: https://doi.org/10.7554/eLife.42182.023
**Figure supplement 3.** Klp9-GFP intensity at anaphase spindles in *ase1^Off* and wild-type cells.
*Figure 6 continued on next page*

*Figure 6 continued*

DOI: https://doi.org/10.7554/eLife.42182.024

**Figure supplement 4.** Klp9-GFP signal length at anaphase spindles in *ase1*$^{Off}$ and wild-type cells.

DOI: https://doi.org/10.7554/eLife.42182.025

Indeed, midzone length, measured by the length of increased mCherry-Atb2 fluorescence at the center of the spindle in a kymograph (*Figure 7—figure supplement 1*), correlated strongly with spindle length (*Figure 7B*: $R^2$ = 0.82, m = 0.26). In *wee1-50* cells the mean midzone length was 2.6 ± 0.4 µm, while it measured 3.2 ± 0.5 µm in wild-type and 5.0 ± 0.6 µm in *cdc25-22* cells. Furthermore, the midzone length, while fluctuating, was kept rather constant throughout the progression of spindle elongation (*Figure 7—figure supplement 2*). This indicates, that the microtubule growth rate is largely correlated to anaphase B spindle elongation.

Furthermore, the number of binding sites for Klp9 on the spindle is not only determined by the overlap length but also by the amount of MTs associated with the midzone. Microtubule density, examined by the mCherry-Atb2 intensity value of a fixed area at the midzone region, slightly increased with spindle length (*Figure 7C*: $R^2$ = 0.53, m = 18.36) whereas values are significantly different between *wee1-50*, wild-type and *cdc25-22* cells (*Figure 7C*). The increased microtubule density of longer spindles is moreover suggested by the fact, that the ratio of tubulin intensity of the anaphase spindle and spindle length, giving the tubulin intensity per unit length of the spindle, scaled with cell size (*Figure 7—figure supplement 3*). Taking both parameters, overlap length and density, into account by measuring the total mCherry-Atb2 intensity of the whole length of the midzone, we observed a strong increase proportional to spindle length (*Figure 7D*: $R^2$ = 0.82, m = 92.22). Thus, longer spindles provide much more binding sites for Klp9 by forming longer regions of antiparallel overlapping microtubules and by bundling more microtubules within this region (*Figure 7E*). Similar to spindle length, the density of the spindle could be regulated by the cytoplasmic volume through limiting the amount of tubulin. This is supported by the fact, that the extent by which the amount of tubulin increases with cell size is greater than the extent by which the spindle length scales with cell size: the ratio of total mCherry-Atb2 intensity and spindle length increases with cell size (*Figure 7—figure supplement 4*). In other words, per unit length of the spindle more tubulin is available in longer cells, potentially resulting in the assembly of more microtubules within the structure.

Interestingly, the ratio of Klp9-GFP intensity and total mCherry-Atb2 intensity of the midzone, which we use as an estimate for the occupancy of the midzone with motors, decreased with spindle length (*Figure 7F*). Hence, within larger cells the midzone of longer spindles might be less occupied with Klp9 molecules in relation to shorter spindles in smaller cells. The inverse correlation of molecular crowding at the spindle midzone might additionally impact the control of spindle elongation velocity. This would explain the difference in the slope of the correlation of spindle elongation velocity and Klp9-GFP intensity between control cells and cells in which Klp9 was overexpressed (*Figure 5—figure supplement 1*). Increasing motor amounts are sufficient to accelerate spindle elongation, but when motors additionally work in a less crowded environment, this might further increase the speed of spindle elongation.

## Discussion

### Mechanism of cell-size-dependent spindle elongation velocity

Our results show that spindle length scales with cell size in fission yeast, and that mitosis duration, defined from spindle assembly to disassembly, is constant irrespective of cell and spindle size (*Figures 1*, *2* and *8*). We reveal that the kinesin-6 Klp9 is an essential player in fixing a constant duration of mitosis in cells of different sizes through adjusting the velocity of spindle elongation in Anaphase B (*Figures 3* and *8*). Central to this conclusion is the observation that deletion of Klp9 eliminates the scaling relationship between spindle elongation velocity and cell size (*Figure 3*). In this background, the mitotic spindle can still elongate, due to the presence of the kinesin-5 Cut7 (*Rincon et al., 2017*), but Cut7 alone does not promote the establishment of a cell-size dependent spindle elongation velocity. Consequently, in absence of Klp9 a correlation between mitosis duration and cell size

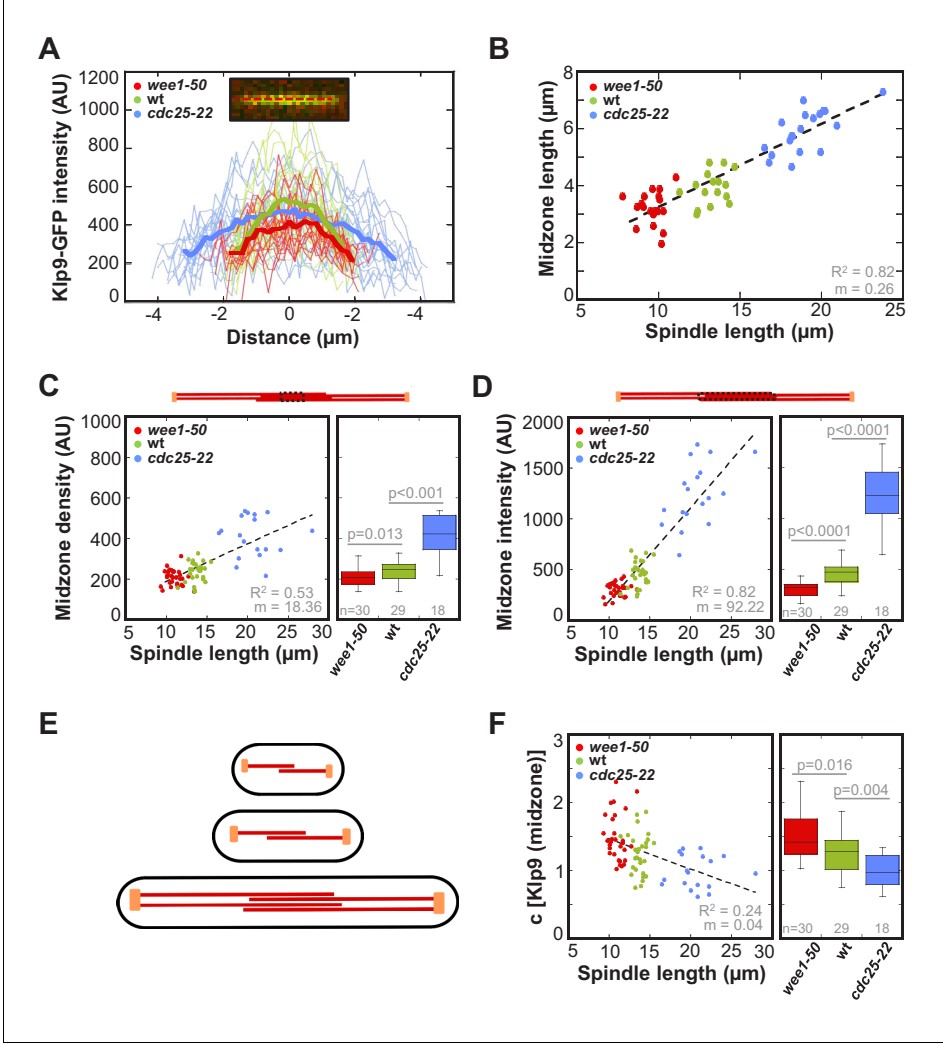

**Figure 7.** Longer spindles provide overproportionally more binding sites for Klp9. (A) Intensities obtained by line scan analysis of Klp9-GFP signals at anaphase spindles of *wee1-50* (n = 20), wild-type (n = 20) and *cdc25-22* (n = 20) cells. A line was placed over the whole length of the Klp9-GFP signal (red dashed line, inset) and the resulting intensity spectrum is shown. Bold curves correspond to the mean intensity distribution for each cell type. (B) Mean midzone length plotted against spindle length (*wee1-50*: n = 19, wt: n = 16, *cdc25-22*: n = 19). (C) mCherry-Atb2 density of the midzone, measured as illustrated by the dashed box within the scheme illustrating the mitotic spindle, plotted against final spindle length (left panel) and box plot comparison of the mCherry-Atb2 density of the midzone (right panel). (D) Total mCherry-Atb2 intensity of the midzone, measured as illustrated by the dashed box within the scheme illustrating the mitotic spindle, plotted against final spindle length (left panel) and box plot comparison of the total mCherry-Atb2 intensity of the midzone (right panel). (E) Model for mitotic spindle structure in fission yeast cells of different sizes. MTs are shown in red and spindle poles in orange. (F) Effective concentration of Klp9 at the midzone plotted against spindle length (left panel) and box plot comparison of the effective concentration of Klp9 at the midzone (right panel). Data in (B–D) and (F) was fitted by linear regression (dashed lines), showing the regression coefficient $R^2$ and the slope m. P values were calculated by Mann-Whitney U test. Data obtained from n analyzed cells was collected from three independent experiments.
DOI: https://doi.org/10.7554/eLife.42182.027

The following figure supplements are available for figure 7:

**Figure supplement 1.** Measurement of midzone length.
DOI: https://doi.org/10.7554/eLife.42182.028

**Figure supplement 2.** Midzone length throughout progressing spindle elongation in anaphase plotted against time.
DOI: https://doi.org/10.7554/eLife.42182.029

*Figure 7 continued on next page*

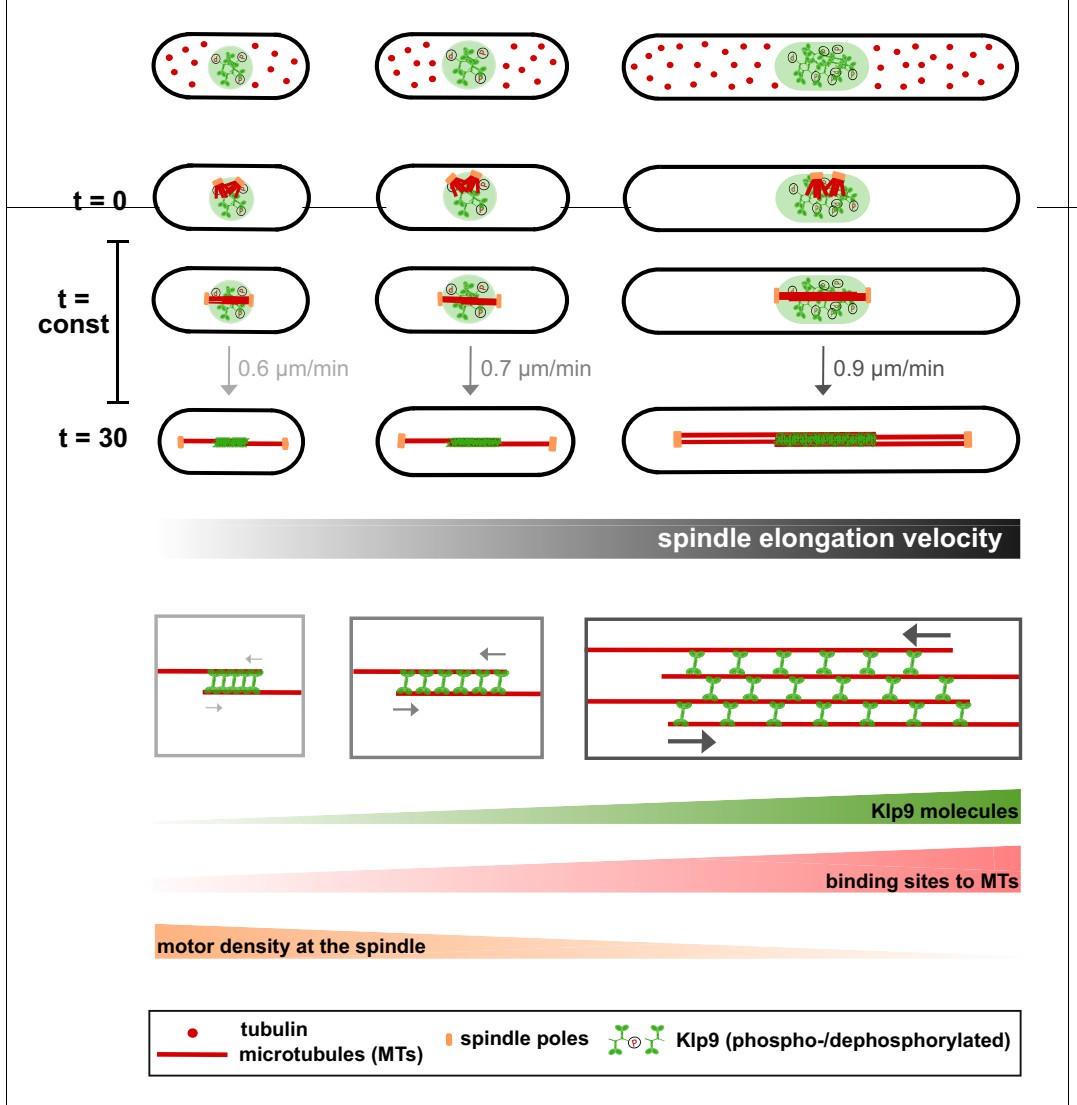

**Figure 8.** Limited components model for the regulation of cell-size-dependent spindle elongation and a constant mitosis duration. Upper panel: The concentration of essential components such as tubulin (red) and Klp9 (green) is constant in cells of different sizes. Thus, total amounts increase proportional to cell size. Due to the presence of more tubulin molecules, longer and more dense spindles can be assembled in larger cells (from t = 0 to t = 30). Likewise, the presence of more Klp9 molecules allows the recruitment of more motors to the anaphase spindle (from t = 0 to t = 30). Lower panel: Since, both the length of antiparallel overlap and the number of microtubules within the midzone increase with cell size, the number of binding sites for Klp9 to the spindle scales overproportionally with spindle and cell length. Thereby, the motor density of Klp9 at the midzone is reduced. With more motors acting in a less crowded environment, spindle elongation can be accelerated and the overall duration of chromosome segregation is kept constant: with increasing cell size, longer spindles are elongated with increasing velocity.

DOI: https://doi.org/10.7554/eLife.42182.032

is established and therefore the constant mitosis duration is abolished. Intensity measurements and overexpression of Klp9 demonstrated that the more Klp9 is available, the more it is recruited to the spindle and the faster the spindle is elongated (*Figures 4*, *5* and *8*).

Klp9 could achieve this role by different mechanisms. First, the forces generated by many individual Klp9 motors might add up and concomitantly elongate the spindle faster than fewer motors. Similarly, the force produced by several full-length *Xenopus* kinesin-5, that crosslink and slide apart two antiparallel microtubules, has been shown to depend linearly on the motor number in vitro

(*Shimamoto et al., 2015*). Why a cell-size-dependent spindle elongation would be established by the kinesin-6 Klp9 but not by the kinesin-5 Cut7 in fission yeast remains an open question. One possibility is that these motors show different behaviors when acting in teams: the forces of several Klp9 motors could add up, but not the forces generated by several Cut7 molecules. Similarly, in contrast to *Xenopus* kinesin-5 (*Shimamoto et al., 2015*), forces generated by kinesin-1 molecules do not scale with motor number in vitro (*Furuta et al., 2013*; *Jamison et al., 2012*). Another possibility is that the force outcome or velocity of Cut7 is regulated by other midzone components such as Klp2 (*Yukawa et al., 2018*) or Ase1, but this may not be the case for Klp9. Such a differential regulation has been reported for the MAP TPX2 and the kinesin-5 Eg5. In vitro TPX2 slowed down MT-sliding of Eg5 but not of kinesin-1 (*Ma et al., 2011*).

Second, the amount of available binding sites for Klp9 to the spindle midzone could be part of the regulation of spindle elongation velocity. Accordingly, the force generated by ensembles of full-length *Xenopus* kinesin-5 scales with the number of motors bound and the overlap length in vitro (*Shimamoto et al., 2015*). Moreover, in vitro studies using the Ase1 homolog PRC1 and the plus-end directed motor Kif4A showed that the sliding velocity of antiparallel oriented microtubules scales with the initial overlap length (*Wijeratne and Subramanian, 2018*). In agreement with this hypothesis we demonstrate, that midzone length scales with cell and spindle length. In addition, we observed that within longer spindles, more microtubules are organized within the midzone (*Figures 7* and *8*). This is consistent with EM-based electron tomographic reconstructions of wild-type *S. pombe* spindles (*Ward et al., 2014*) and serial-section reconstructions of *cdc25-22* spindles (*Ding et al., 1993*). At the metaphase-to-anaphase transition, spindles in wild-type cells consist of about nine microtubules and *cdc25-22* spindles contain approximately sixteen microtubules (*Ding et al., 1993*; *Ward et al., 2014*). With both overlap length and density correlating with cell size, the number of Klp9 binding sites to microtubules increases to a greater extent than the number of Klp9 molecules at the midzone. This results in a lower effective concentration of the motor at the midzone of longer spindles (*Figures 7F* and *8*). We speculate that this local concentration could assist the regulation of a cell-size-dependent elongation speed. Indeed, increasing motor densities, for instance of kinesin-8, have been reported to reduce motor velocity (*Leduc et al., 2012*). As motors entered a region of high motor density, their velocity was slowed down, potentially due to steric interference. This phenomenon might result from the fact that the motor cannot step forward if the next binding-site is already occupied by another motor (*Leduc et al., 2012*). Similarly, increasing densities of Cut7, immobilized on a coverslip, decreased the sliding velocity of attached microtubules (*Britto et al., 2016*).

Third, the velocity of growth of antiparallel microtubules organized in the spindle midzone could impact the speed of spindle elongation. The fact that the overlap length did not shrink during spindle elongation, suggests that the associated microtubules grow constantly, as reported for Ptk1 cells and diatom spindles (*Cande, 1986*; *Cande and McDonald, 1985*; *Masuda and Cande, 1987*; *Saxton and McIntosh, 1987*). One could thus think that, instead of the velocity of microtubule sliding, the velocity of microtubule growth could determine the velocity of spindle elongation. Accordingly, in *Drosophila* embryos depletion of the kinesin-8 KLP67A, which is thought to trigger microtubule plus-end depolymerization (*Varga et al., 2009*; *Varga et al., 2006*), increased the velocity of microtubule sliding and thus of anaphase B spindle elongation (*Wang et al., 2010*). If this would similarly be the case in fission yeast, we think, that Klp9 would be part of the regulation of microtubule growth, since the scaling relationship of spindle elongation velocity and cell size was completely abolished in absence of Klp9. Moreover, increasing the number of Klp9 motors increased the velocity of spindle elongation, which would not occur if the speed of microtubule growth determines the speed of microtubule sliding independent of Klp9. The kinesin-6 could achieve this role by either acting as a MT polymerase itself, as it has been suggested for the other MT sliding motor kinesin-5 (*Chen and Hancock, 2015*) or by recruiting other spindle components that regulate microtubule polymerization. In contrast to this model, the velocity of microtubule sliding could impact the velocity of microtubule growth. In this case, the velocity of microtubule sliding determines the velocity of spindle elongation, while the rate of microtubule growth is adjusted accordingly, preventing the overlap region to shrink throughout spindle elongation. An example for this model, proposing microtubule growth to follow microtubule sliding, was established by in vitro studies using PRC1 and the kinesin-4 Xklp1, which is recruited to overlap regions by PRC1 and inhibits microtubule growth (*Bieling et al., 2010*). The number of kinesin-4 molecules is determined by the overlap length. Thus,

upon decreasing overlap length due to microtubule sliding, the number of kinesin-4 molecules may decrease, enabling the overlap to grow. Consequently, microtubule polymerization follows the rate of microtubule sliding. In general, it is not clear which parameter eventually determines the dynamics of spindle elongation and this could be different from one organism to another. In fission yeast, reducing the microtubule growth rate by disrupting the interaction between Mal3, an EB1 protein, and Dis1, a member of the XMAP215/TOG family, resulted in a slightly faster spindle elongation (*Matsuo et al., 2016*). This result is hard to interpret and one would expect the following two outcomes. Either, in case the microtubule growth rate is limiting, spindle elongation velocity would decrease; or, if the microtubule sliding velocity is limiting for the speed of spindle elongation, spindle elongation velocity would be unaltered. However, decreasing the speed of microtubule growth does not seem to directly result in a decreased velocity of spindle elongation and thus might not be limiting in fission yeast.

To conclude, we propose a model in which larger cells provide more Klp9 molecules resulting in the recruitment of more motors to the spindle midzone (*Figure 8*). An increased amount of motors at the midzone accelerates the velocity of spindle elongation, either by increasing the velocity of microtubule sliding or by increasing the rate of microtubule growth. In addition, this effect may be enhanced by a lower rate of molecular crowding on longer spindles (*Figure 8*: lower panel).

Cumulatively, this study suggests that cell-size-dependent spindle elongation velocity, which ultimately ensures a constant mitotic time frame in cells of various sizes, may be a consequence of scaling of essential spindle components or parameters with cell size: the number of Klp9 molecules, microtubule density and midzone length (*Figure 8*). We thus wonder if, like spindle length (*Good et al., 2013*; *Hazel et al., 2013*), the constant mitotic time frame could be regulated by the cytoplasmic volume. Consistent with this model we found the concentration of tubulin and Klp9 to be constant in fission yeast cells of different sizes. With an unaltered concentration, total amounts increase proportionally to cell size: larger cells provide more tubulin and Klp9 molecules (*Figure 8*). This eventually leads to the assembly of longer and more dense mitotic spindles, the recruitment of more motors to these spindles, and consequently an acceleration of spindle elongation.

## Constant mitosis duration

With this study we describe one mechanism that allows diverse cells to accomplish chromosome segregation in a constant time frame. It was previously reported that in various human cell lines the duration of G1-, S- and G2-phase varied widely while mitosis duration was remarkably constant (*Araujo et al., 2016*). Together, this sheds light on the importance of the regulation of mitosis duration and provides evidence that chromosome segregation has to be finished within a certain time frame. This time frame might even be similar in different organisms. The time needed to accomplish chromosome segregation in mammalian cells and in yeast is strikingly similar. Mammalian cells spend 20–60 min in mitosis (*Yang et al., 2008*) with an average of 45 min in various human cell lines (*Araujo et al., 2016*). Fission yeast spends approximately 30 min in mitosis. In contrast, interphase duration accounts for 12–30 hr in mammalian cells (*Ganem and Pellman, 2012*; *Yang et al., 2008*), but only for around 3 hr in *S. pombe* cells.

## Materials and methods

### Production of *S. pombe* mutant strains

All used strains are isogenic to wild-type 972 and were obtained from genetic crosses, selected by random spore germination and replica on plates with according drugs or supplements. All strains are listed in the *Supplementary file 1*. Gene deletion and tagging was performed as described previously (*Bähler et al., 1998*).

### Fission yeast culture

All *S. pombe* strains were maintained at 25°C and cultured in standard media. Strains were either maintained on YE5S plates or EMM plates additionnally containing adenine, leucine and uracil in case of the Ase1-Shut-Off strain, at 25°C and refreshed every third day. Generally, one day before imaging cells were transferred into liquid YE5S and imaged the following day at exponential growth.

Cells in starvation phase were grown until an optical density at a wavelength of 600nm ($OD_{600}$) of 1.2–1.5 and then imaged. To generate abnormally long cells by treatment of wild-type cells with Hydroxyurea (HU, sigma aldrich), 110 mM HU was added to the cell culture (in liquid YE5S) at exponential growth. After incubation of 3 to 4 hr at 25°C, cells were washed three times with $H_2O$ and resuspended in YE5S. Following one hour at 25 °C cells were imaged.

The *cdc2-asM17* mutants were arrested by adding 1 μM NM-PP1 (Calbiochem, CA) to the culture at exponential growth. After 30 min, 2 hr or 2.5 hr of incubation at 25°C, cells were washed three times with $H_2O$ and resuspended in YE5S. Following one hour at 25 °C cells were imaged.

For overexpression experiments, strains (pnmt-klp9, pnmt-GFP-klp9 and pnmt-ase1) were cultured two days before imaging in liquid EMM with adenine, leucine, uracil and thiamine. The next day cells were centrifuged at 3000 rpm for 5 min, washed three times with $H_2O$ and resuspended in EMM with adenine, leucine and uracil. After 18 to 20 hr at 25 °C cells were imaged.

For shut-off experiments, strains (pnmt81-ase1) were cultured in EMM with adenine, leucine and uracil two days before imaging. Approximately 20 hr before imaging, cells were transferred into YE5S.

## Live microscopy

For live-cell imaging cells were mounted on YE5S agarose pads, containing 2% agarose (*Tran et al., 2004*). Imaging was performed at 25°C.

Images were acquired on a motorized inverted Nikon Eclipse Ti-E microscope, equipped with a spinning-disk CSUX1 confocal head (Yokogawa Electric Corporation), a Plan Apochromat 100x/1.45 N.A. oil immersion objective lens (Nikon), a PIFOC objective stepper, a Mad City Lab piezo stage, a CCD CoolSNAP HQ2 camera (Photometrics) and a laser bench (Erol) with 491 and 561 nm diode lasers, 100 mW each (Cobolt). Stacks of 7 planes spaced by 1 μm were acquired for each channel with 400 msec exposure time, binning two and an electronic gain of 3 for both wavelengths.

For each time-lapse movie, an image was taken every minute for a duration of 90–120 min.

## Image analysis

Using Metamorph 7.2, maximum projections of each stack were performed for analysis of spindle dynamics and for presentation and sum projections for intensity measurements.

Spindle dynamics were examined by the length of the mCherry-Atb2 signal over time. Metaphase and anaphase spindle length refers to the maximum spindle length of each phase.

Intensity measurements were performed by drawing a region around the area of interest, reading out the average intensity per pixel, subtracting the background and multiplying this value with the size of the area.

Kymographs for the determination of midzone length were constructed with Metamorph.

## Quantification and statistical tests

All graphs and box plots were generated using Kaleidagraph 4.5 (Synergy). Analysis of statistical significance was performed by using Mann-Whitney U test, which unlike the Student T-test does not assume a normal distribution of values. Obtained p values are shown within box plots and data sets were defined as significantly different if $p > 0.05$. N values, representing the number of cells, are displayed either in the figure legend or inside the box plot. Data was fitted by linear regression analysis and the obtained coefficient of determination ($R^2$) and the slope (m) of the corresponding regression line is displayed inside the graph. Within boxplots, the box encloses 50% of each data set with the median value displayed as a line. Top or bottom of the box correspond to the value halfway between median and the largest or smallest data value. Lines extending from top and bottom of a box illustrate the maximum or minimum values of the data set, which fall within an acceptable range. Values outside this range are displayed as individual points.

## Acknowledgements

We thank Sergio Rincon and Marcel Hennes for fruitful discussions and critical reading of the manuscript. We appreciate the maintenance of microscopes by Vincent Fraisier and Ludovic Leconte from PICT- IBiSA Imaging facility (Institut Curie), a member of the France-BioImaging national research infrastructure, where imaging was performed. LKK is supported by a PhD fellowship from the French

Ministry of Science and Education. This work was supported in part by grants from La Ligue, Fondation ARC, and INCa to PTT. We thank the lab of Masamitsu Sato (Waseda University) for generously providing strains.

## Additional information

### Funding

| Funder | Grant reference number | Author |
|---|---|---|
| Fondation ARC pour la Recherche sur le Cancer | | Phong Thanh Tran |
| Institut National Du Cancer | | Phong Thanh Tran |
| Ligue Contre le Cancer | Ile de France | Phong Thanh Tran |

The funders had no role in study design, data collection and interpretation, or the decision to submit the work for publication.

### Author contributions

Lara Katharina Krüger, Conceptualization, Formal analysis, Investigation, Visualization, Methodology, Writing—original draft, Writing—review and editing; Jérémie-Luc Sanchez, Data curation, Investigation; Anne Paoletti, Phong Thanh Tran, Supervision, Funding acquisition, Writing—review and editing

### Author ORCIDs

Lara Katharina Krüger https://orcid.org/0000-0002-0439-951X
Phong Thanh Tran http://orcid.org/0000-0002-2410-2277

### Decision letter and Author response

Decision letter https://doi.org/10.7554/eLife.42182.036
Author response https://doi.org/10.7554/eLife.42182.037

## Additional files

### Supplementary files

• Supplementary file 1. *S. pombe* strain list.
DOI: https://doi.org/10.7554/eLife.42182.033

• Transparent reporting form
DOI: https://doi.org/10.7554/eLife.42182.034

### Data availability

All data are included in the manuscript.

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
