## [Decision Letter]

Thank you for submitting your article "Kinesin-6 regulates cell-size-dependent spindle elongation velocity to keep mitosis duration constant in fission yeast" for consideration by *eLife*. Your article has been reviewed by three peer reviewers, one of whom is a member of our Board of Reviewing Editors, and the evaluation has been overseen by Andrea Musacchio as the Senior Editor. The reviewers have opted to remain anonymous.

The reviewers have discussed the reviews with one another and the Reviewing Editor has drafted this decision to help you prepare a revised submission.

Summary:

The reviewers agree that this manuscript provides novel and interesting information addressing an important scaling question. The reviewers agree that the majority of the conclusions are well supported by quantitative measurements of high quality. The main novel findings are: (i) In fission yeast, spindle length and elongation rate scale with cell size, (ii) the increase in spindle elongation rate with cell length keeps the mitosis duration constant in cells of different sizes, (iii) Klp9 regulates spindle elongation rate, which increases with motor number, keeping mitosis duration in fission yeast the same in cells of different size. The main concern of the reviewers was that they felt that the molecular mechanism controlling the recruitment of Klp9 to the spindle and consequently controlling elongation rate was not completely clear. Particularly, a major concern was that the observed scaling effects might not necessarily be a consequence of cell size, but rather of differences in the basic *cdc2* activity levels in the mutant cells used for this study. It was felt that this concern should be addressed experimentally to support the statement that the effects reported here depend indeed on cell size and not differences in basic *cdc2* activity levels. In conclusion, we would be happy to consider a revised version of the manuscript addressing the reviewers' concerns as outlined below.

Essential revisions:

1) Missing control experiments:

The authors should provide data indicating that Klp9 recruitment to the spindle midzone also correlates with spindle length/cell size in cells without manipulation of the basic levels of *cdc2* activity. Klp9 is regulated by *cdc2* activity (four phospho-sites) as shown in Meadows et al. Cell Rep. 2017 – thus, comparing cells with different basic *cdc2* activity levels is a concern. The authors can overcome this *cdc2* activity problem by using for example the *cdc2-asM17* allele to acutely control *cdc2* activity with ATP analogues (Aoi et al., 2014). This strain would allow them to arrest the cells for various times to make them longer and release them into mitosis by washing out the inhibitor.

2) Technical concern requiring clarification:

Figure 4B. The graph gives Klp9-GFP intensity versus cell length. Anaphase B is from 3-15 µm. At what spindle length did the authors measure Klp9-GFP intensity? End of anaphase B? Ideally, the authors should compare for all three cell types the same spindle length (e.g. 5 µm) and end of anaphase.

3) Scholarship:

The authors state, "a strong correlation of spindle length and size exists in *S. pombe* demonstrating that this process is not restricted to embryogenesis". A similar statement is also in the Abstract. However, it has been previously shown in budding yeast and human cells that spindle length scales with cell length (Rizk et al., 2014; Yang et al., 2016). Previous literature should be cited and the statement adjusted accordingly.

4) Presentation of the model:

The authors could probably improve the description of their model of the scaling mechanism. In particular, they could better integrate the idea of the 'thickness' of midzone bundles playing a role in the scaling mechanism. This observation is reported rather late, and appears to motivate a last-minute U-turn of the reasoning. In the first part of the manuscript a model seems to be developed mainly based on 'more motors producing more force'. Towards the end of the manuscript the model is apparently also based on the idea of 'a lower motor density in bundles being somehow better for faster speed'. The Abstract however states as a main concept 'limited components' being the basis of the scaling model, which does seem to be something different.

---

## [Author Response]

Essential revisions:1) Missing control experiments:The authors should provide data indicating that Klp9 recruitment to the spindle midzone also correlates with spindle length/cell size in cells without manipulation of the basic levels of cdc2 activity. Klp9 is regulated by cdc2 activity (four phospho-sites) as shown in Meadows et al. Cell Rep. 2017 – thus, comparing cells with different basic cdc2 activity levels is a concern. The authors can overcome this cdc2 activity problem by using for example the cdc2-asM17 allele to acutely control cdc2 activity with ATP analogues (Aoi et al., 2014). This strain would allow them to arrest the cells for various times to make them longer and release them into mitosis by washing out the inhibitor.

We agree with the reviewers. To show that the observed effect of an increased Klp9 recruitment and the consequent acceleration of spindle elongation results from changes in cell size and not from the mutants itself, we performed the suggested experiment, using the *cdc2-asM17* mutant. Using this mutant, we could confirm that with increasing cell length, more motors are recruited to the longer spindles and that the velocity of spindle elongation is increased. Moreover, we used two additional methods to alter cell size: wild type cells in starvation (short) and wild type cells treated with hydroxyurea (long). In these cells, the velocity of spindle elongation scaled similarly with cell size as in the other mutants, indicating that the observed effects indeed account for changes in cell size. These new experiments are in complete agreement with our previous results, and are presented in the new Figure 2.

2) Technical concern requiring clarification:Figure 4B. The graph gives Klp9-GFP intensity versus cell length. Anaphase B is from 3-15 µm. At what spindle length did the authors measure Klp9-GFP intensity? End of anaphase B? Ideally, the authors should compare for all three cell types the same spindle length (e.g. 5 µm) and end of anaphase.

To overcome the lack of clarity concerning the Klp9-GFP intensity measurements, we added the following experiment. The intensity of the Klp9-GFP intensity was followed over time throughout the whole phase of spindle elongation. Thereby, it can be clearly observed that the intensity of Klp9-GFP at the midzone does not change dramatically throughout spindle elongation and that, most importantly, the intensity increases with cell size at every given time point of spindle elongation. The new measurements are shown in the new Figure 4.

3) Scholarship:The authors state, "a strong correlation of spindle length and size exists in S. pombe demonstrating that this process is not restricted to embryogenesis". A similar statement is also in the Abstract. However, it has been previously shown in budding yeast and human cells that spindle length scales with cell length (Rizk et al., 2014; Yang et al., 2016). Previous literature should be cited and the statement adjusted accordingly.

We thank the reviewers for pointing out these papers. We missed them previously. They have now been included in the revision.

4) Presentation of the model:The authors could probably improve the description of their model of the scaling mechanism. In particular, they could better integrate the idea of the 'thickness' of midzone bundles playing a role in the scaling mechanism. This observation is reported rather late, and appears to motivate a last-minute U-turn of the reasoning. In the first part of the manuscript a model seems to be developed mainly based on 'more motors producing more force'. Towards the end of the manuscript the model is apparently also based on the idea of 'a lower motor density in bundles being somehow better for faster speed'. The Abstract however states as a main concept 'limited components' being the basis of the scaling model, which does seem to be something different.

We agree that the hypothesis of molecular crowding appeared as a “last-minute U-turn” and that the strong increase of Klp9 binding sites to microtubules may not be a straightforward example for a limited components model. However, we think that also the density of the mitotic spindle, as well as spindle length, can be determined by the available amount of tubulin, as stated in the paragraph “Microtubule density and overlap length increase with spindle length”. To improve the model, we tried to give this hypothesis of more binding sites an equal importance to the hypothesis of the motor number controlling the speed of spindle elongation in the Discussion.